# Evaluation of the LDBio *Aspergillus* ICT lateral flow assay for serodiagnosis of allergic bronchopulmonary aspergillosis

**Elizabeth Stucky Hunter**[1], **Iain D. Page**[1,2], **Malcolm D. Richardson**[1,3], **David W. Denning**[1,4]*

**1** Division of Infection, Immunity and Respiratory Medicine, Faculty of Biology, Medicine and Health, Manchester Academic Health Science Centre, University of Manchester, Manchester, United Kingdom, **2** North Manchester General Hospital, Pennine Acute Hospitals NHS Trust, Manchester, United Kingdom, **3** Mycology Reference Centre Manchester, Manchester University NHS Foundation Trust, Manchester, United Kingdom, **4** National Aspergillosis Centre, Manchester University NHS Foundation Trust, Manchester, United Kingdom

\* ddenning@manchester.ac.uk

**Data Availability Statement:** All relevant data are within the paper and supporting information files.

## Abstract

### Background

Early recognition and diagnosis of allergic bronchopulmonary aspergillosis (ABPA) is critical to improve patient symptoms, and antifungal therapy may prevent or delay progression of bronchiectasis and development of chronic pulmonary aspergillosis.

### Objective

A recently commercialized lateral flow assay (*Aspergillus* ICT) (LDBio Diagnostics, Lyons, France) detects *Aspergillus*-specific antibodies in <30 minutes, requiring minimal laboratory equipment. We evaluated this assay for diagnosis of ABPA compared to diseased (asthma and/or bronchiectasis) controls.

### Methods

ABPA and control sera collected at the National Aspergillosis Centre (Manchester, UK) and/ or from the Manchester Allergy, Respiratory and Thoracic Surgery research biobank were evaluated using the *Aspergillus* ICT assay. Results were read both visually and digitally (using a lateral flow reader). Serological *Aspergillus*-specific IgG and IgE, and total IgE titres were measured by ImmunoCAP.

### Results

For 106 cases of ABPA versus all diseased controls, sensitivity and specificity for the *Aspergillus* ICT were 90.6% and 87.2%, respectively. Sensitivity for 'proven' ABPA alone (n = 96) was 89.8%, and 94.4% for 'presumed' ABPA (n = 18). 'Asthma only' controls (no bronchiectasis) and 'bronchiectasis controls' exhibited 91.4% and 81.7% specificity, respectively. Comparison of *Aspergillus* ICT result with *Aspergillus*-specific IgG and IgE titres showed no

**Funding:** This work was supported by the Medical Research Council (Newton grant MR/P017622/1 and the NIHR Manchester Biomedical Research Centre, both to DWD.

**Competing interests:** Dr. Denning and family hold Founder shares in F2G Ltd, a University of Manchester spin-out antifungal discovery company. He acts or has recently acted as a consultant to Scynexis, Cidara, Pulmatrix, Zambon, Pulmocide, iCo Therapeutics, Roivant, Biosergen, Mayne Pharma, Bright Angel Therapeutics and Fujifilm. In the last 3 years, he has been paid for talks on behalf of Dynamiker, Hikma, Gilead, Merck, Mylan and Pfizer. He is a longstanding member of the Infectious Disease Society of America Aspergillosis Guidelines group, the European Society for Clinical Microbiology and Infectious Diseases Aspergillosis Guidelines group and the British Society for Medical Mycology Standards of Care committee. Dr. Richardson is the co-founder of Richardson Bio-Tech (Guangzhou) Ltd, acts as a consultant for Gilead Sciences, Pfizer Inc., MSD, Mylan, and gives paid for presentations on behalf of these companies. He is a member of the joint European Confederation for Medical Mycology and European Society for Clinical Mycology and Infectious Diseases Guidelines writing group.

**Abbreviations:** ABPA, allergic bronchopulmonary aspergillosis; ABPA-B, allergic bronchopulmonary aspergillosis with bronchiectasis; ABPA-S, serological allergic bronchopulmonary aspergillosis (no bronchiectasis evident; CPA, chronic pulmonary aspergillosis; CF, cystic fibrosis; COPD, chronic obstructive pulmonary disease; DOR, diagnostic odds ratio; EIA, enzyme immunoassay; ELISA, enzyme-linked immunosorbent assay; ICT, immunochromatographic technology; kU$_A$/ml, kilo-units antigen-specific antibodies/millilitre (ImmunoCAP; ManARTS, Manchester Allergy, Respiratory, and Thoracic Surgery; mgA/L, milligrams antigen-specific antibodies/liter (ImmunoCAP); NAC, National Aspergillosis Centre.

evident immunoglobulin isotype bias. Digital measurements displayed no correlation between ImmunoCAP *Aspergillus*-specific IgE level and ICT test line intensity.

## Conclusions

The *Aspergillus* ICT assay exhibits good sensitivity for ABPA serological screening. It is easy to perform and interpret, using minimal equipment and resources; and provides a valuable simple screening resource to rapidly distinguish more serious respiratory conditions from *Aspergillus* sensitization alone.

## Introduction

Allergic bronchopulmonary aspergillosis (ABPA) is an immunologically-mediated pulmonary disorder caused by hypersensitivity to the allergens of *Aspergillus* species (predominantly *A. fumigatus*) [1, 2]. The majority of cases complicate asthma—representing an estimated 1–4% of adult asthma cases worldwide [3]—but ABPA can also complicate cystic fibrosis (CF) [4] and in rarer instances, can occur in patients with neither condition [5, 6]. ABPA usually manifests clinically as poorly controlled asthma often with recurrent pulmonary infections, production of thick mucus plugs, and/or fatigue [7]. It remains an under-diagnosed disease in many settings [8, 9], and delays in diagnosis result in an increased likelihood of disease progression and permanent lung damage through the development of bronchiectasis and/or chronic pulmonary aspergillosis (CPA) if left untreated [1]. There are thought to be over 4.5 million adults with ABPA across the world, assuming about 2.5% of adults with asthma are affected [3]; ABPA is rare in children.

Recently revised diagnostic criteria for ABPA [10] relies primarily on serology, including the major requirements of raised total IgE and evidence of *Aspergillus* sensitivity by raised *Aspergillus*-specific (*Asp*) IgE or skin prick testing. In addition, two minor criteria must also be present and may include radiological features consistent with ABPA, eosinophilia, or raised levels of *Asp* IgG. A review comparing serological tests for identifying ABPA found that assays for total and *Asp* IgE demonstrated good sensitivity but poor to moderate specificity (40–80%), whereas the precipitins assay for the detection of *Asp* IgG (& IgA) [11, 12] had high specificity but poor sensitivity [13]. Other serological assays for the detection of *Asp* IgG are commercially available, such as enzyme-linked immunosorbent assay (ELISA)/enzyme immunoassay (EIA) [14], but their performance is dependent on cut-off values that vary between regional populations as well as for the specific condition being diagnosed [12].

LDBio Diagnostics (Lyons, France) has introduced a new point-of-care lateral flow assay (*Aspergillus* IgG-IgM ICT) for detection of *Asp* antibodies. The assay utilizes immunochromatographic technology (ICT) and has been validated against a spectrum of *Aspergillus*-related diseases [15], including a moderate number (n = 74) of ABPA cases. We recently evaluated patients with CPA and found this assay to have a sensitivity of 90.6% and a specificity of 98.0% [16]. Here we assess its performance in the serological diagnosis of ABPA.

## Materials and methods

### Serological samples

This study was performed using serum samples collected from 113 ABPA patients, using a combination of convenience sampling from patients identified at the National Aspergillosis

Centre (NAC) (Manchester, UK) (n = 67) and samples archived in the Manchester Allergy, Respiratory and Thoracic Surgery (ManARTS) (Manchester, UK) research tissue bank (n = 46). The NAC is a nationally commissioned service providing long-term specialist care for patients with aspergillosis throughout the UK. For convenience sampling, residual sera from patient samples collected as part of routine care between 06/2018 and 01/2019 were identified. Control serum samples obtained from ManARTS were collected under ethical consent between 2011–2016 approved by the Local Research Ethics Committee (LREC, ethics approval 15/NW.0409). All sera were stored at -80˚C until use. This study was assessed through the NHS Health Research Authority system (HRA) (http://www.hra-decisiontools.org.uk/research/) and was found to meet the UK NHS definition of a retrospective service evaluation for which formal ethical review was therefore not required.

Each ABPA diagnosis was confirmed by an experienced specialist clinician and classified as 'proven' (n = 95) or 'presumed' (n = 18). Using ISHAM guidelines [10, 11], the following three criteria were required for a 'proven' ABPA diagnosis: (1) total IgE >1000 IU/ml, (2) positive *Aspergillus*-specific IgE or skin prick test, and (3) any two of the following 'other' criteria: [a] positive *Aspergillus*-specific IgG or precipitins, [b] eosinophils >500 cells/µl, [c] radiological features consistent with ABPA. A 'presumed' ABPA diagnosis required (2) and (3) as above, with total IgE between 113–999 IU/ml. ImmunoCAP EIA (Thermo Scientific, Waltham, MA) was used for measurement of total IgE, *Aspergillus*-specific IgE (positive result >0.35 $kU_A$/ml [kilo-units antigen-specific antibodies/milliliter] IgE), and *Aspergillus*-specific IgG (positive result >40mgA/L [milligrams antigen-specific antibodies/liter] IgG) levels. The Ouchterlony method [11] was used to detect *Aspergillus* precipitins (Microgen Bioproducts, Surrey, UK). Within the set of 'proven' ABPA cases, seven (7) had ABPA complicating cystic fibrosis and were analysed separately.

Diseased controls were obtained from the ManARTS biobank, a research tissue bank comprised of samples from consenting patients with respiratory and allergic diseases (and other related conditions) and patients undergoing thoracic surgery, at the Northwest Lung Research Centre at Wythenshawe Hospital (Manchester University NHS Trust Foundation). Serum samples from 164 patients with bronchiectasis and/or asthma were identified using two sets of inclusion criteria. 'Asthma' control cases (n = 93) were defined as asthmatic patients with a normal chest X-ray and total IgE <1000 IU/ml within one year of sample collection; and with no clinical or diagnostic evidence of CPA, ABPA, bronchiectasis or any other lung diseases that might predispose to CPA including COPD, sarcoidosis, prior tuberculosis, cancer, and rheumatoid arthritis. Patients with *Aspergillus* sensitivity, allergic rhinitis/nasal polyps, severe food or drug allergies, or anaphylaxis were also excluded from this control group. 'Bronchiectasis' control cases (n = 71) included patients with computed tomography evidence of bronchiectasis. Patients with a diagnosis of ABPA, CPA, or *Aspergillus* bronchitis, or signs of ABPA or CPA on radiology were excluded. Bronchiectasis cases with underlying asthma (n = 48) and/or COPD (n = 10) were included, as well as patients with microbiological (respiratory culture) evidence of *Aspergillus* colonization (n = 3) or raised *Aspergillus*-specific IgE (> 0.35 $kU_A$/ml IgE) (n = 5).

## Serological analysis

Each sample was tested using the *Aspergillus* ICT IgG-IgM (LDBio Diagnostics, Lyon, France) lateral flow assay. Test kits were shipped at ambient temperature and stored at 4˚C upon receipt. Each batch (10/pack) of ICT cartridges were equilibrated to ambient laboratory temperature before use and were run according to the manufacturer's instructions. Briefly, 15µl of sera was dispensed onto the ICT sample application pad, followed by four drops of eluting

solution (provided with kit). The test was read at 30 minutes and results were interpreted both visually (by 'eye') and digitally using the Qiagen ESEQuant LR3 (Lake Constance, Germany) lateral flow reader. Both reads were conducted by the same user, with the visual reading conducted first to eliminate bias resulting from the digital reading. For visual reads, the test was determined positive by the appearance of two lines: a blue positive control ("C") line, and a black positive test ("T") line. The appearance of any black line at the "T" marker was considered positive, as recommended in the manufacturer's guidelines. Using the LR3 lateral flow reader, a positive test was defined by detection of peaks (any height) between 46.0–48.0mm and between 53.5–55.5mm for control and test lines, respectively.

### Routine diagnostics

In many cases, *Aspergillus*-specific IgG and IgE, and total IgE levels were measured on ABPA patient samples as part of routine clinical care. Testing was carried out by the Manchester University NHS Foundation Trust, Department of Immunology, using the automated Immuno-CAP Phadia 1000 System.

### Statistical analyses

To assess diagnostic performance, we calculated Youden's J statistic (sensitivity + specificity–1), likelihood ratios and the diagnostic odds ratio (DOR) [17]. Binomial confidence intervals (95%) were calculated for sensitivity, specificity, and DOR. Pearson's chi-square statistic was used for comparisons between subgroups. Spearman's rank correlation coefficient ($\rho$) was used for correlation between ImmunoCAP titers and ICT results. For all results, a two-tailed *P*-value < 0.05 was considered statistically significant.

## Results

### Patients and sera

Patient characteristics for 106 ABPA patients and 141 diseased controls are shown in Table 1. Within the ABPA patient group, two subgroups were defined as 'proven' or 'presumed' according to total serum IgE level ('proven': >1000 IU/ml, 'presumed': 113–999 IU/ml). Diseased controls consisted of patients diagnosed with asthma and/or bronchiectasis. Subgroups for analysis were defined as (1) ABPA-S: serological ABPA (no bronchiectasis evident, ± asthma), and (2) ABPA-B: ABPA or control with bronchiectasis ± any other conditions (including asthma and COPD). Additionally, as part of routine testing, 64 of the 106 ABPA patients in this study had microbiological culture performed on respiratory samples (sputum), usually multiple specimens. High volume fungal culture technique was usually used, which has a higher yield than routine culture [18]. Seven (7) cases yielded no growth. *Aspergillus fumigatus* was the main pathogen isolated in the remaining 57 cases, and 21 of these also grew other *Aspergillus* species in culture. The following data was not available for 'asthma only' control subjects: (1) gender, (2) whether sputum culture was performed (and results).

### ICT results

For all 106 ABPA serum samples, 96 tested positive by ICT with 90.6% sensitivity (95% CI, 83.3% to 95.4%). Sensitivity for 'proven' and 'presumed' ABPA subgroups was 89.8% (95% CI, 81.5% to 95.2%) and 94.4% (95% CI, 72.7% to 99.9%), respectively, with no significant difference between these groups. No associations with any of multiple parameters were linked with false negative assay ICT assay. In the total diseased control group, 143 of the 164 sera tested negative by ICT with 87.2% specificity (95% CI, 81.1% to 91.9%) (Table 2).

**Table 1. Patient and control characteristics.**

| Characteristic | ABPA | | | Controls | | |
|---|---|---|---|---|---|---|
| | All[a] (n = 106) | Proven (n = 88) | Presumed (n = 18) | All[b] (n = 164) | Asthma (n = 93) | Bronchiectasis (n = 71) |
| Female gender, n (%) | **46 (43.4)** | 36 (40.9) | 10 (55.6) | **n/a[c]** | n/a | 43 (60.6) |
| Mean age (years) | **63** | 63 | 67 | **52** | 45 | 60 |
| Age range (years) | **18–90** | 18–90 | 52–83 | **16–87** | 16–68 | 22–87 |
| Asthma, n (%) | **93 (87.7)** | 78 (88.6) | 15 (83.3) | **141 (86.0)** | 93 (100) | 48 (67.6) |
| Bronchiectasis, n (%) | **82 (76.4)** | 66 (75.0) | 16 (88.9) | **71 (43.3)** | 0 (0) | 71 (100) |
| Asthma + bronchiectasis, n (%) | **72 (67.9)** | 59 (67.0) | 13 (72.2) | **48 (29.3)** | 0 (0) | 48 (67.6) |
| COPD[d], n (%) | **9 (8.5)** | 8 (9.1) | 1 (5.6) | **7 (4.3)** | 0 (0) | 10 (14.1) |
| Cystic fibrosis (CF), n (%) | **0 (0)** | 0 (0) | 0 (0) | **0 (0)** | 0 (0) | 0 (0) |
| *Aspergillus fumigatus* growth in sputum culture, n (%) | **57 (53.8)** | 48 (54.5) | 9 (50.0) | 3 (1.9) | 0 (0) | 3 (4.2) |
| *A. fumigatus* only, n (%) | **36 (34.0)** | 28 (31.8) | 8 (44.4) | n/a | 0 (0) | n/a |
| *A. fumigatus* + other *Aspergillus* spp., n (%) | **21[e] (19.8)** | 20 (22.7) | 1 (5.6) | n/a | 0 (0) | n/a |
| Other *Aspergillus* spp. (only) growth in sputum culture, n (%) | **0 (0)** | 0 (0) | 0 (0) | n/a | 0 (0) | n/a |

[a]All ABPA: proven + presumed

[b]All control: asthma ± bronchiectasis

[c]n/a = data not available

[d]COPD = chronic obstructive pulmonary disease.

[e]*A. niger* [17], *A. nidulans* [3], *A. fischeri* [1], *A. montevidensis* [1], *A. terreus* [1], *A. versicolor* [1], *A. pallidoflavus* [1].

Analysis of subgroups based on underlying conditions demonstrated a significant difference in sensitivity between the ABPA-B and ABPA-S subgroups (93.9% and 79.2% sensitivity, respectively; Pearson's chi-square statistic = 4.719, $P = 0.03$). No significant differences in specificity were observed. For three (3) cases of bronchiectasis with evidence of *Aspergillus* colonization by positive sputum culture results, there were no ICT positive results (100% specificity) and for five (5) cases of bronchiectasis with raised *Asp* IgE, two (2) cases were ICT positive (60% specificity). We also evaluated serum from seven (7) cystic fibrosis patients with 'proven' ABPA and found 100% sensitivity using the ICT for this group. Youden's J statistic calculated for overall ICT test results indicated a good balance between sensitivity and specificity. The test had an overall positive likelihood ratio of 7.07 (95% CI, 4.72 to 10.59) and negative likelihood ratio of 0.11 (95% CI, 0.06 to 0.20), with a high DOR of 65 (95% CI, 30 to 145). All tests were read both visually (i.e., by eye) and digitally (using the Qiagen ESEQuant LR3 lateral flow reader), with 100% agreement between methods.

**Table 2. Summary of results for LDBio *Aspergillus* ICT IgG-IgM test and routine serological assay.**

| Analysis group | | Sensitivity | | Specificity | | Youden's | DOR |
|---|---|---|---|---|---|---|---|
| | | % | (95% CI) | % | 95% CI (%) | Index | (95% CI) |
| **Overall** | **All ABPA (n = 106)** | **90.6** | **(83.3–95.4)** | | | **0.816** | **65 (30–145)** |
| | 'Proven' ABPA (n = 88) | 89.8 | (81.5–95.2) | | | | |
| | 'Presumed' ABPA (n = 18) | 94.4 | (72.7–99.9) | | | | |
| | **All diseased controls (n = 164)** | | | **87.2** | **(81.1–91.9)** | | |
| ABPA-S (Serological ABPA) | Serological ABPA (n = 24) | 79.2 | (57.9–92.9) | | | 0.706 | 62 (19–203) |
| | Asthma only controls (n = 93) | | | 91.4 | (83.8–96.2) | | |
| ABPA-B (ABPA + bronchiectasis) | ABPA + any bronchiectasis (n = 82) | 93.9 | (86.3–98.0) | | | 0.750 | 69 (23–204) |
| | Bronchiectasis controls (n = 71) | | | 81.7 | (70.7–89.9) | | |

**Table 3. LDBio *Aspergillus* ICT performance in ABPA cases with *Aspergillus fumigatus* and non-*A. fumigatus* species.**

| Sputum culture result (*n* = 64) | *n* | ICT + (*n*) | % sensitivity (95% CI) |
|---|---|---|---|
| **All *Aspergillus* growth (culture positive)** | **57** | **55** | **96.5 (87.9, 99.6)** |
| A. fumigatus only | 36 | 35 | 97.2 (85.5, 99.9) |
| A. fumigatus + other Aspergillus spp. | 21 | 20 | 95.2 (76.2, 99.9) |
| A. niger | 17 | 16 | 94.1 (71.3, 99.9) |
| A. nidulans | 3 | 3 | 100 (29.2, 100) |
| A. fischeri | 1 | 1 | 100 (2.5, 100) |
| A. montevidensis | 1 | 1 | 100 (2.5, 100) |
| A. pallidoflavus | 1 | 1 | 100 (2.5, 100) |
| A. terreus | 1 | 1 | 100 (2.5, 100) |
| A. versicolor | 1 | 1 | 100 (2.5, 100) |
| **No *Aspergillus* growth (culture negative)** | **7** | **7** | **100 (59, 100)** |

In relation to *Aspergillus* species detected in sputum from ABPA patients (Table 3), the ICT performed well in detecting *Asp* antibodies for cases with *A. fumigatus* isolated alone (97.2% sensitivity) as well as for cases where non-*A. fumigatus* species were isolated in addition (95.2% sensitivity), with no significant difference (Pearson's chi-square statistic = 0.154; $P$ = 0.69). There was no significant difference in sensitivity between samples from patients with at least one *Aspergillus* species isolated by culture ('culture positive', 96.5% sensitivity) and patients whose sputum yielded no culture growth ('culture negative', 100% sensitivity). Of the 42 patients for whom no sputum culture was conducted, 34 were positive by ICT (81.0% sensitivity; 95% CI, 65.9% to 91.4%).

Finally, we assessed three (3) different production lots of the LDBio *Aspergillus* ICT assay kits using 30 ABPA serum samples and observed 100% agreement of results across all lots tested (data not shown).

## ICT detection of immunoglobulin isotypes

The *Aspergillus* ICT uses a homogeneous antigen sandwich format to detect *Asp* antibody in patient sera (Fig 1A), and may theoretically detect immunoglobulin isotypes other than (as claimed by the manufacturer) IgG and IgM. For 52 ABPA samples in this study, ImmunoCAP *Asp* IgE and IgG results were available and compared to the ICT result. We observed no evident immunoglobulin isotype bias for the ICT assay (Fig 1B), but did observe a significant correlation between *Asp* IgG and IgE titers (Spearman's rank correlation coefficient $\rho$ = 0.3524, $P$ = 0.01).

Using data generated by digitally reading the ICT on the Qiagen LR3 reader, we found a weak but significant correlation between endpoint (30 minutes) test line peak height and ImmunoCAP *Asp* IgG titer (n = 46, Spearman's rank correlation coefficient $\rho$ = 0.2961, $P$ = 0.046), supporting our previous findings[16]. However, as with the previous findings[16], the correlation was not sufficient for quantification of *Asp* IgG. We found no significant correlation between peak height and ImmunoCAP total IgE (n = 48, Spearman's rank correlation coefficient $\rho$ = 0.2101, $P$ = 0.15) or *Asp* IgE (n = 48, Spearman's rank correlation coefficient $\rho$ = 0.0295, $P$ = 0.84) titers.

## Discussion

Global prevalence of ABPA is estimated at 1–4% of adult asthma patients (approximately 4.8 million people) worldwide [3], but regional estimates may be higher. A study in Rio de Janiero

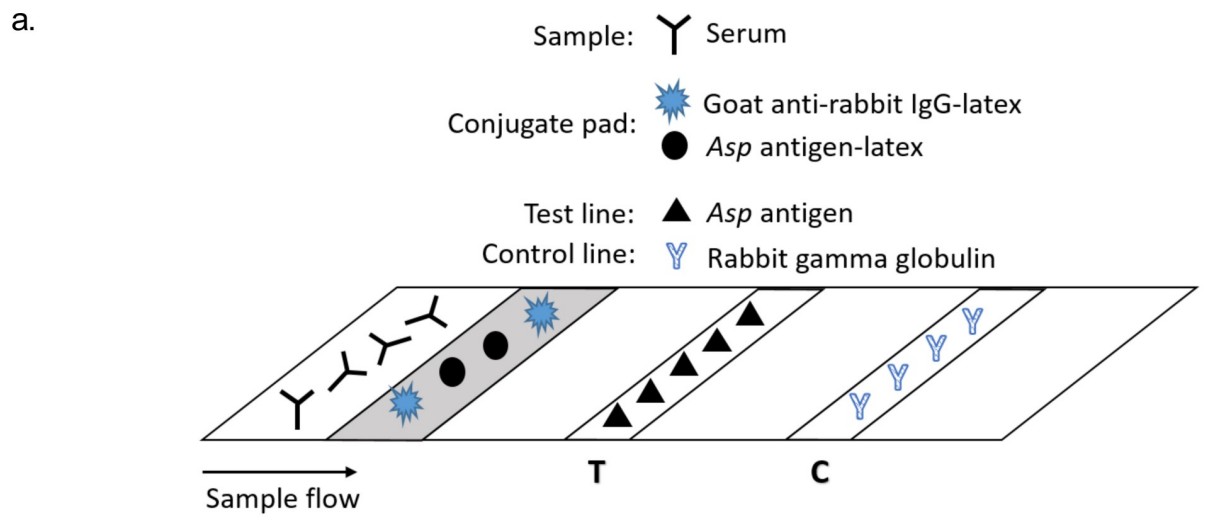

**Fig 1.** (a) LDBio *Aspergillus* ICT assay format and (b) Comparison of LDBio *Aspergillus* ICT result with ImmunoCAP *Asp* IgG and IgE titers (n = 52). Positive result cut-off for each assay denoted by dashed line (ImmunoCAP *Aspergillus* IgG = 40mg/L, ImmunoCAP *Aspergillus* IgE = 0.35 kU$_A$/L).

found 20% prevalence among Brazilian asthmatic patients [19], and several studies in India found ABPA in approximately 5% [20] to 8% [5, 21], and up to 20% [22] of asthmatics, with up to half of patients being misdiagnosed as pulmonary tuberculosis [9]. The incidence of ABPA is also higher in patients with severe asthma and/or corticosteroid-dependent asthma (7–14%), and in patients with atopy [23]. Diagnosis of ABPA relies on a body of clinical, radiological, and immunological (and/or mycological) evidence; and the degree of difficulty in diagnosing ABPA largely depends on the staging and severity of disease. Diagnosis is relatively straightforward when a patient exhibits all clinical symptoms including bronchiectasis, and most patients are diagnosed at this stage [24]. Earlier diagnosis, however, is ideal to prevent development of bronchiectasis and irreversible tissue damage [25]. Serological tests are a useful tool for early detection of ABPA and comprise a significant portion of the recently suggested guidelines for ABPA diagnosis [10], with raised total IgE and *Asp* IgE (or positive skin prick test) being 'major' criteria and raised *Asp* IgG or positive precipitins test included under 'minor' criteria.

The LDBio *Aspergillus* ICT lateral flow assay is a new diagnostic test requiring minimal time and resources. Our recent evaluation in CPA patients (vs. healthy controls) determined the test to have good sensitivity (91.6%) and specificity (98.0%), and a significant improvement in performance over our current workhorse assay (ImmunoCAP EIA) to detect raised *Asp* IgG [16]. The current evaluation in ABPA patients and diseased controls from the United Kingdom has shown the *Aspergillus* ICT to have good overall sensitivity (90.6%) and specificity (87.2%) in distinguishing patients with ABPA from those with underlying respiratory disease. Not surprisingly, the test exhibited significantly better sensitivity for ABPA with bronchiectasis (ABPA-B) (93.9%) than for serological ABPA (ABPA-S) ± asthma (no evidence of bronchiectasis) (79.2%). Bronchiectasis is associated with progression of ABPA and a more severe form of disease [1]. Persistent *Aspergillus* infection may lead directly to bronchiectasis or drive recurrent exacerbations and progression of bronchiectasis already present due to ABPA or severe asthma [26, 27]. Once present, the pathophysiology of bronchiectasis can facilitate persistent infection through impairment in mucociliary clearance [28] and structural/tissue damage that creates a permissive environment for establishment of infection [26, 29, 30]. Patients with bronchiectasis (with or without ABPA) may develop *Aspergillus* bronchitis, and this entity is also associated with raised *Asp* IgG antibodies [31]. We did not explicitly assess a group of asthmatic patients with *Aspergillus* sensitisation for *Aspergillus* IgG with the ICT or ImmunoCap, and given the likely interaction with *Aspergillus* bronchitis, we would need to do so knowing if *Aspergillus* bronchitis was or was not present.

All cases in this study with *Aspergillus* species growth in high-volume sputum culture were positive for *Aspergillus fumigatus* and approximately one-third (37%) were also positive for other *Aspergillus* species. There was no significant difference in ICT performance between cases with *A. fumigatus* growth alone versus those with other species present. There were no cases in this study with non-*A. fumigatus* ABPA, however, a limited number of non-*A. fumigatus* CPA cases were evaluated in a previous study and we found no significant difference in ICT performance between *A. fumigatus* and non-*A. fumigatus* CPA cases [16]. The ability to detect antibody to non-*A. fumigatus* may be of particular importance in regions where ABPA due to non-*A. fumigatus* species is more prevalent [32]. For the diagnosis of ABPA, sputum culture is considered a supportive (not diagnostic) test [10]. In Indian patients thought to have ABPA, sputum tested by microscopy and culture was positive for *Aspergillus* species in only 32.5% of 203 cases [8]. Higher volume cultures yielded a positive in 56% of 75 cases (similar to this series), some of whom were on antifungal therapy compared with conventional culture (12.5%) [18]. *Aspergillus* ICT was more sensitive than this. Culture has the advantage of being

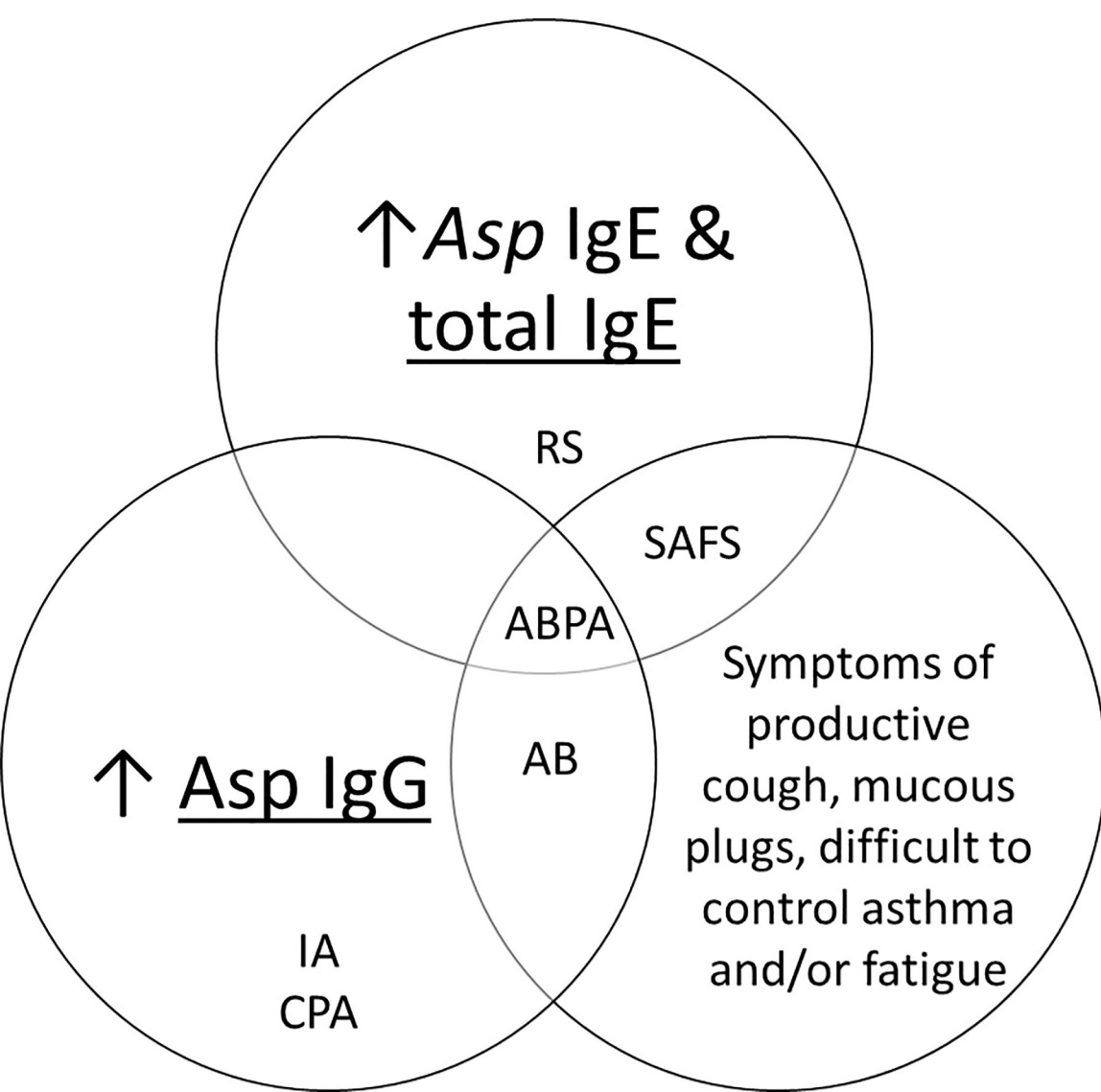

**Fig 2. Interpretation of *Aspergillus* serology and clinical presentation of ABPA for diagnosing *Aspergillus* disease: allergic bronchopulmonary aspergillosis (ABPA), *Aspergillus* rhinosinusitis (RS), severe asthma with fungal sensitization (SAFS), *Aspergillus* bronchitis (AB), invasive aspergillosis (IA), and chronic pulmonary aspergillosis (CPA).**

able to check antifungal susceptibility, if positive, although direct detection of resistance is now possible [33–36].

Given the 'major' and 'minor' requirements of both raised *Asp* IgE and IgG for ABPA diagnosis, we sought to determine if the homogeneous antigen-sandwich format of the *Aspergillus* ICT rendered it capable of detecting IgE (in addition to IgG and IgM as claimed) and whether the test displayed an isotype bias. Although we found a significant correlation between levels of *Asp* IgG and *Asp* IgE, we did not observe any such isotype bias using the ICT. As expected based on our previous findings [16], we found the test to perform well at increasing levels of *Asp* IgG (ImmunoCAP 'positive', >40mgA/L) regardless of *Asp* IgE titer. The test also performed well at raised levels of *Asp* IgE paired with values of *Asp* IgG considered 'negative' by the recommended ImmunoCAP cut-off ($\leq$ 40mgA/L). While this may possibly indicate the

detection of *Asp* IgE by the ICT test format, we have previously found this assay to detect *Asp* antibody in patients with clinically confirmed CPA who have tested 'negative' by ImmunoCAP *Asp* IgG [16, 37]. Further studies would be necessary to evaluate the individual contributions of IgG and IgE from patient sera and the antibody class-specific performance of the *Aspergillus* ICT.

Individual serological tests alone for total IgE or *Asp* IgE or IgG are not specific for ABPA. It is important to note that routine tests used for ABPA diagnosis—raised *Asp* IgE, IgG and/or positive precipitins test—are also used in the diagnostic pathways for *Aspergillus* diseases other than ABPA including *Aspergillus* bronchitis (AB), acute and sub-acute invasive aspergillosis (IA), CPA, severe asthma with fungal sensitization (SAFS), and chronic or granulomatous *Aspergillus* rhinosinusitis (RS) [12]. The diagnostic interpretation of these tests, alone or in combination with each other and/or clinical presentation is summarized in Fig 2. Under generally accepted diagnostic criteria for *Aspergillus* disease, ABPA is the only condition using the combined requirements of raised *Asp* IgG and IgE. In practice however, it is not uncommon to see raised total and/or *Asp* IgE in CPA patients [38], especially those that have developed CPA as a result of untreated ABPA progression [39]. Additionally, although a raised level of *Asp* IgG is considered an exclusion criteria for SAFS, approximately 10% of all asthmatics present with raised *Asp* IgG [1] and there are likely to be cases of SAFS with increased levels of both *Asp* IgG and IgE, but lacking the clinical and radiological features of ABPA [12]. High titers of total serum IgE are encountered not only in ABPA, but also in cases of SAFS, *Aspergillus* rhinosinusitis, and in atopic asthmatics [40, 41]. Furthermore, cut-off values to distinguish these conditions are speculative and may even be different in ABPA complicating asthma versus ABPA complicating CF [4, 42].

In this study of clinically confirmed cases of ABPA compared to diseased controls, we found the LDBio *Aspergillus* ICT to have good sensitivity and specificity. The test effectively distinguished between *Aspergillus*-sensitization complicating asthma and/or bronchiectasis, and underlying conditions. It is rapid (result in <30 minutes) and easy to perform, with simple result interpretation by visible inspection. Overall, the LDBio *Aspergillus* ICT exhibits excellent performance as a screening tool in the ABPA diagnostic pathway.

## Supporting information

**S1 Dataset.**
(XLSX)

## Acknowledgments

Diseased control sera were provided by the ManARTS research tissue bank, Wythenshawe Hospital, Manchester University NHS Foundation Trust (Manchester, UK).

## Author Contributions

**Conceptualization:** Malcolm D. Richardson, David W. Denning.

**Data curation:** Iain D. Page.

**Formal analysis:** Elizabeth Stucky Hunter.

**Funding acquisition:** David W. Denning.

**Investigation:** Elizabeth Stucky Hunter, David W. Denning.

**Methodology:** Elizabeth Stucky Hunter.

**Project administration:** Malcolm D. Richardson, David W. Denning.

**Resources:** Iain D. Page.

**Supervision:** Malcolm D. Richardson, David W. Denning.

**Visualization:** Elizabeth Stucky Hunter.

**Writing – original draft:** Elizabeth Stucky Hunter.

**Writing – review & editing:** Iain D. Page, Malcolm D. Richardson, David W. Denning.

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
