## [Decision Letter · Decision Letter 0]

30 Apr 2020

PONE-D-20-06967

Evaluation of the LDBio Aspergillus ICT lateral flow assay for serodiagnosis of allergic bronchopulmonary aspergillosis

PLOS ONE

Dear Dr. Denning,

Thank you for submitting your manuscript to PLOS ONE. After careful consideration, we feel that it has merit but does not fully meet PLOS ONE’s publication criteria as it currently stands. Therefore, we invite you to submit a revised version of the manuscript that addresses the points raised during the review process.

In particular, some more details about the patients should be added, the classification of patient subgroups have to be clarified and the discussion needs some revision.

We would appreciate receiving your revised manuscript by Jun 14 2020 11:59PM. To enhance the reproducibility of your results, we recommend that if applicable you deposit your laboratory protocols in protocols.io, where a protocol can be assigned its own identifier (DOI) such that it can be cited independently in the future. For instructions see: http://journals.plos.org/plosone/s/submission-guidelines#loc-laboratory-protocols

We look forward to receiving your revised manuscript.

Kind regards,

Olaf Kniemeyer

Academic Editor

PLOS ONE

2. In your Methods section, please provide additional information regarding the residual sera used in the study. Please specify the source of the residual sera, whether it was collected and used with the approval of an IRB, and whether you had consent from patients or next of kin.

4. Thank you for stating the following in the CONFLICT OF INTEREST STATEMENT of your manuscript:

"Dr. Denning and family hold Founder shares in F2G Ltd, a University of Manchester spin-out

antifungal discovery company. He acts or has recently acted as a consultant to Scynexis, Cidara,

Pulmatrix, Zambon, Pulmocide, iCo Therapeutics, Roivant, Biosergen, Mayne Pharma and Fujifilm.

In the last 3 years, he has been paid for talks on behalf of Dynamiker, Hikma, Gilead, Merck, Mylan

and Pfizer. He is a longstanding member of the Infectious Disease Society of America Aspergillosis

Guidelines group, the European Society for Clinical Microbiology and Infectious Diseases

Aspergillosis Guidelines group and the British Society for Medical Mycology Standards of Care

committee. Dr. Richardson acts as a consultant for Gilead Sciences, Pfizer Inc., MSD, Mylan, and

gives paid for presentations on behalf of these companies. He is a member of the joint European

Confederation for Medical Mycology and European Society for Clinical Mycology and Infectious

Diseases Guidelines writing group."

Reviewers' comments:

Reviewer's Responses to Questions

**Comments to the Author**

1. Is the manuscript technically sound, and do the data support the conclusions?

Reviewer #1: Yes

Reviewer #2: Yes

2. Has the statistical analysis been performed appropriately and rigorously? 

Reviewer #1: Yes

Reviewer #2: Yes

3. Have the authors made all data underlying the findings in their manuscript fully available?

Reviewer #1: Yes

Reviewer #2: Yes

4. Is the manuscript presented in an intelligible fashion and written in standard English?

Reviewer #1: Yes

Reviewer #2: Yes

5. Review Comments to the Author

Reviewer #1: I read with great interest the well written manuscript by Stucky Hunter et al. entitled: “Evaluation of the LDBio Aspergillus ICT lateral flow assay for serodiagnosis of allergic bronchopulmonary aspergillosis”.

This study presents the diagnostic indices of a rapid and easy to perform lateral flow assay for the diagnosis of ABPA. The topic is of major interest because APBPA diagnosis is particularly challenging.

The study design and methodology are adequate. The results are well presented and discussed. Overall, the LDBio Aspergillus ICT exhibited excellent performance for ABPA screening.

Yet, the authors should detail the characteristics of the patients – especially those with a “proven” ABPA -- who displayed a false negative results with this assay.

Reviewer #2: The authors describe the results of LDBio Aspergillus ICT lateral flow assay in ABPA. It is an excellent paper from a group who are world leaders in Aspergillus research. I have few comments which will further improve the manuscript:

1. The subgroups for analysis are very confusing. It should be restricted to ABPA-B and ABPA-S.

2. Also, the Youden’s index and DOR is of no value in this analysis and should be removed.

3. It would have been interesting if the authors had A.fumigatus-IgG in all cases. Then using a predefined cutoff, they could have compared the diagnostic performance of LDBio vs. Phadia. This point can be discussed.

4. The discussion is unusually long and may be shortened.

6. PLOS authors have the option to publish the peer review history of their article (what does this mean?). If published, this will include your full peer review and any attached files.

Reviewer #1: Yes: Stéphane Ranque

Reviewer #2: No

---

## [Author Response · Author response to Decision Letter 0]

4 Aug 2020

Joerg Heber, Editor-in-Chief

PLoS One

PLOS

Carlyle House

Carlyle Road

Cambridge, CB4 3DN

United Kingdom

July 17, 2020

Dr. Heber,

Thank you very much for the opportunity to revise and resubmit Manuscript ID PONE-D-20-06967, entitled “Evaluation of the LDBio Aspergillus ICT lateral flow assay for serodiagnosis of allergic bronchopulmonary aspergillosis” for consideration for publication in PLoS One. The authors greatly appreciate the comments of the reviewers and believe they have helped increase the quality of the manuscript. 

Please find included in our submission the revised manuscript (including the marked-up document), as well as the original decision letter with our responses to the reviewers found directly under each comment in red (“Response to Reviewers). We have made several changes to the manuscript to fully address the reviewers’ comments and to clarify our objectives, methods, and results.

With regards to data availability and sharing of a de-identified data set, we have uploaded an anlymised dataset. 

We have additionally updated the Conflict of Interest statement within the manuscript to include the required phrasing and it now reads:

“Dr. Denning and family hold Founder shares in F2G Ltd, a University of Manchester spin-out antifungal discovery company. He acts or has recently acted as a consultant to Scynexis, Cidara, Pulmatrix, Zambon, Pulmocide, iCo Therapeutics, Roivant, Biosergen, Mayne Pharma and Fujifilm. In the last 3 years, he has been paid for talks on behalf of Dynamiker, Hikma, Gilead, Merck, Mylan and Pfizer. He is a longstanding member of the Infectious Disease Society of America Aspergillosis Guidelines group, the European Society for Clinical Microbiology and Infectious Diseases Aspergillosis Guidelines group and the British Society for Medical Mycology Standards of Care committee. Dr. Richardson acts as a consultant for Gilead Sciences, Pfizer Inc., MSD, Mylan, and gives paid for presentations on behalf of these companies. He is a member of the joint European Confederation for Medical Mycology and European Society for Clinical Mycology and Infectious Diseases Guidelines writing group. This does not alter our adherence to PLOS ONE policies on sharing data and materials.”

Also, as requested, we have amended our methods section to provide additional information regarding the residual sera used in the study, to include ethical approval details for control samples and the NHS Health Research Authority system (HRA) assessment determining this study to be a retrospective service evaluation for which ethical evaluation is not required. 

Regarding reference to ‘Data not shown’ for the results described on lines 224-226 (in marked up document), we have removed this descriptor. The evaluation and analysis of these 7 samples for the cystic fibrosis subgroup is described in the methods section (lines 157-158) and the results detailed in the respective sections. There is no other data regarding this subgroup that is not described and/or shown in the manuscript.

We look forward to your decision, and thank you for considering our manuscript for publication in PLoS One. 

Sincerely,

D W Denning FRCP FRCPath DCH FIDSA FMedSCi

Professor of Infectious Diseases in Global Health

Reviewer's Responses to Questions

Comments to the Author

5. Review Comments to the Author

Reviewer #1: I read with great interest the well written manuscript by Stucky Hunter et al. entitled: “Evaluation of the LDBio Aspergillus ICT lateral flow assay for serodiagnosis of allergic bronchopulmonary aspergillosis”. This study presents the diagnostic indices of a rapid and easy to perform lateral flow assay for the diagnosis of ABPA. The topic is of major interest because APBPA diagnosis is particularly challenging. The study design and methodology are adequate. The results are well presented and discussed. Overall, the LDBio Aspergillus ICT exhibited excellent performance for ABPA screening. Yet, the authors should detail the characteristics of the patients – especially those with a “proven” ABPA -- who displayed a false negative results with this assay.

Thank you, this is indeed an important consideration. To explore this, we have compared LDBio results with the presence or absence of the following parameters for patients in the “proven” ABPA group:

(1) Asp IgG >40

(2) Raised eosinophils

(3) Positive ppt test

(4) Positive PCR test

(5) Positive Culture

(6) Characteristic radiology

(7) Asthma

(8) Bronchiectasis

(9) COPD

And have found no significant correlation between false negative (or otherwise) results and 

these parameters. 

Reviewer #2: The authors describe the results of LDBio Aspergillus ICT lateral flow assay in ABPA. It is an excellent paper from a group who are world leaders in Aspergillus research. I have few comments which will further improve the manuscript:

1. The subgroups for analysis are very confusing. It should be restricted to ABPA-B and ABPA-S.

2. Also, the Youden’s index and DOR is of no value in this analysis and should be removed.

3. It would have been interesting if the authors had A.fumigatus-IgG in all cases. Then using a predefined cutoff, they could have compared the diagnostic performance of LDBio vs. Phadia. This point can be discussed.

4. The discussion is unusually long and may be shortened.

Thank you for your feedback. To address these points: 

(1) We have amended our methods and patient characteristic sections to remove subgroups other than ABPA-S and ABPA-B. Figure 1 has been removed as a result, and Table 2 (and associated text) has been updated to reflect this.

(2) We feel that Youden’s statistic and DOR are important statistics to compare assay performance between ABPA patient subgroups (using matched controls for each subgroup in the calculation of these statistics). 

(3) As this study used retrospectively collected patient samples to assess performance of the LDBio assay, obtaining A. fumigatus IgG was beyond the scope of this study. Data for A. fumigatus IgG was obtained only if the assay was run as part of routine clinical care. This was not the case for all patient samples assayed.

(4) The discussion has been edited to reduce length, to the extent possible.

---

## [Decision Letter · Decision Letter 1]

26 Aug 2020

Evaluation of the LDBio Aspergillus ICT lateral flow assay for serodiagnosis of allergic bronchopulmonary aspergillosis

PONE-D-20-06967R1

Dear Dr. Denning,

We’re pleased to inform you that your manuscript has been judged scientifically suitable for publication and will be formally accepted for publication once it meets all outstanding technical requirements.

Kind regards,

Olaf Kniemeyer

Academic Editor

PLOS ONE

Additional Editor Comments (optional):

Reviewers' comments:

Reviewer's Responses to Questions

**Comments to the Author**

1. If the authors have adequately addressed your comments raised in a previous round of review and you feel that this manuscript is now acceptable for publication, you may indicate that here to bypass the “Comments to the Author” section, enter your conflict of interest statement in the “Confidential to Editor” section, and submit your "Accept" recommendation.

Reviewer #2: All comments have been addressed

2. Is the manuscript technically sound, and do the data support the conclusions?

Reviewer #2: Yes

3. Has the statistical analysis been performed appropriately and rigorously? 

Reviewer #2: Yes

4. Have the authors made all data underlying the findings in their manuscript fully available?

Reviewer #2: Yes

5. Is the manuscript presented in an intelligible fashion and written in standard English?

Reviewer #2: Yes

6. Review Comments to the Author

Reviewer #2: -

7. PLOS authors have the option to publish the peer review history of their article (what does this mean?). If published, this will include your full peer review and any attached files.

Reviewer #2: **Yes: **Ritesh Agarwal

---

## [Editor Report · Acceptance letter]

16 Sep 2020

PONE-D-20-06967R1 

Evaluation of the LDBio Aspergillus ICT lateral flow assay for serodiagnosis of allergic bronchopulmonary aspergillosis 

Dear Dr. Denning:

I'm pleased to inform you that your manuscript has been deemed suitable for publication in PLOS ONE. Congratulations! Your manuscript is now with our production department. 

Kind regards, 

on behalf of

Dr. Olaf Kniemeyer 

Academic Editor

PLOS ONE